# Objective Assessment of Perilymphatic Fistula in Cases of Postoperative Vertigo after Cochlear Implantation by Cochlin Tomoprotein (CTP)

**DOI:** 10.3390/brainsci13111525

**Published:** 2023-10-29

**Authors:** Ingo Todt, Tetsuo Ikezono

**Affiliations:** 1Department of Otolaryngology, Head and Neck Surgery, Bielefeld University, Medical Faculty OWL, Campus Klinikum Bielefeld, Teutoburgerstr. 50, 33604 Bielefeld, Germany; 2Department of Otolaryngology, Head and Neck Surgery, Saitama University, Irumagun, Saitamaken 350-0495, Japan

**Keywords:** cochlin tomoprotein (CTP), cochlear implantation (CI), perilymphatic fistula (PLF), vertigo, dizziness

## Abstract

Objective: Vertigo is a quite frequent complication after cochlear implantation. Perilymphatic fistula (PLF) is assumed to be one cause of this problem. Cochlin tomoprotein (CTP) is a newly introduced marker for PLF. The present aim was to evaluate the rate of positive CTP testing in cases of newly occurring vertigo after cochlear implantation. Materials and Methods: Twelve patients with vertigo after cochlear implantation and a revisional electrode-sealing procedure underwent intraoperative rinsing of their middle ear. The sample was evaluated for CTP with monoclonal antibody testing. Sixteen controls from six CI patients were taken. Results: 4 out of 12 (33%) cases showed positive CTP testing, indicating that a PLF could be evaluated. In all of the positive CTP cases, surgery decreased the vertigo symptoms. A relation between the subjective visual assessment of a fistula and a positive CTP value was not observed. Controls confirmed the value of the testing. Discussion: CTP detection objectively shows that PLF can occur in patients with vertigo after CI.

## 1. Introduction

Cochlear implantation (CI) is the standard of choice for auditory rehabilitation in cases of severe-to-profound hearing loss. The surgical procedure has a low risk of complications. However, postoperative vertigo is a well-known and quite frequent complication after cochlear implantation. The published rate for new occurrence of vertigo after cochlear implantation varies from 13% to 74% [1,2]. In addition to timely and variable occurrence, various reasons are known as causes of vertigo and dizziness after cochlear implantation. Possible reasons include mechanical fluid pressure trauma; vestibular receptor affection [3]; electrical co-stimulation [4]; changes occurring in the inner ear after CI, which can take the form of endolymphatic hydrops [5]; and otoconial dislocation. Furthermore, foreign body reactions, an intraoperative loss of perilymph, labyrinthitis, and endolymphatic hydrops [6] can be assumed to be causative.

A perilymphatic fistula (PLF) is defined as a loss of perilymph at the oval or round window and is known as a rare complication of stapes surgery, head trauma, and barotraumatic trauma [7,8]. Sealing of a PLF is a successful treatment option for some cases of sudden sensorineural hearing loss and vertigo [9]. A recent study showed that electrode resealing is a successful tool in cases of vertigo after cochlear implantation [10]. Different clinical tests have been introduced to observe PLF, including fistula tests and intraoperative pressure transmission tests, but they have not spread into clinical routines due to lack of evidence, results, and clinical outcomes after round/oval window occlusion. Based on subjective tests, a rate of PLF between 40% and 7% was estimated among patients with SSNHL [11,12,13]. Publications have shown that the intraoperative visual observation was not correlated with the clinical postoperative outcome in terms of increased hearing threshold after occlusion [14].

Objective assessment of PLF by collecting the well-introduced β-trace testing is technically difficult due to the lack of specificity and the small amount of fluid [15].

Ikezono and coworkers enabled new access to objective observations of perilymphatic fistulas by detecting a cochlea-specific protein called cochlin tomoprotein (CTP) with a monoclonal antibody test [16]. It was found to be stable and detectable in small amounts [17]. Clinically, perilymphatic fistula after stapes surgery and penetrating middle ear trauma have been detected with this method [18,19]. Perilymph leakage in minor middle/inner ear anomaly cases was identified by CTP test, imaging study, and intraoperative identifications [20,21].

The aim of this study was to observe the occurrence of middle ear CTP values, as an indicator of PLF, in cases of new occurrences of vertigo after cochlear implantation. Additionally, a control group was used to evaluate the values from testing of PLF.

## 2. Material and Methods

### 2.1. Sample Collection

To detect perilymph leakage in the middle ear, we used samples collected by lavaging the middle ear cavity with 0.3 mL of saline and recovering the fluid, defined as middle ear lavage (MEL). The MEL was centrifuged (6000 rpm for 15 s; Eppendorf Systems, Hamburg, Germany), and the supernatant was collected and stored at −20 °C. Afterward, it was put on dry ice for further testing of CTP. Probes were blinded to the observing laboratory. The cut-off values for monoclonal AB ELISA CTP were as follows: <30 ng/mL was no PLF, 30–60 ng/mL was intermediate PLF, and >60 ng/mL was sure PLF (CTP ELISA: TECAN/IBL:301170068; at the time of testing this was not commercially available).

### 2.2. Subjects

Study group: We included 12 CI cases which visited our hearing ambulance. ELISA analysis was performed between 2021 and 2023. Mean age was 54.5 y, with 6 female and 6 male implantees. The number of CI surgeries during that period was 221 in the first authors-based clinic. The total number of revisions for this period was 18. Revisions were performed as follows: endaural incision, elevation of a tympanomeatal flap, identification of the electrode and taking a middle ear fluid sample, removal of the tissue patch around the electrode, placing a new tissue patch around the electrode and in the round window, sealing the middle ear against the inner ear space.

The criteria for the tympanotomy and resealing of the electrode were as follows:(1)No vestibular symptom preoperatively;(2)The new occurrence of vertigo after CI surgery:
(A)That still persisted unchanged after 4 days of steroid and antibiotic medication;(B)Which was so bothersome that the patient asked for advice. Resealing was offered as a treatment option;(C)After a vertigo-free period.

All patients undergoing revisions fully consented to a CTP detection test. Exclusion criteria were age under 12 years and preoperative vertigo before CI surgery.

In one case, revision surgery (No. 2) was performed after two days since the patient woke up after surgery with a persisting nystagmus. All patients showed no classic clinical features of a PLF.

### 2.3. Other Outcome and Measures

Intraoperatively, the surgeon assumed a PL leakage by visualization under the microscope. A visual-based assumption of PLF was made if the middle ear mucosa and the old fascia seal at the round window area were hyperplastic with a wet surface. Even a stapes movement was tested to cause a fluid leakage. After an endaural approach without irrigation, a resealing was performed with fascia and fibrin glue after removal of the old fascia seal from around the electrode and the round window was performed. Three weeks postoperatively, patients were interviewed as to whether they still had vertigo/dizziness or whether there had been an improvement in the symptoms. The nature of the vestibular symptoms was expressed as rotatory vertigo or unsteadiness.

### 2.4. Control Group

Additionally, we studied 6 control patients during regular cochlear implant surgeries to verify the correlation between monoclonal CTP ELISA testing and clinical expected results with 6 CI cases (Table 2). Control group patients were without vertigo.

MEL was collected in each of the 4 conditions during CI surgery, with a set of samples (A, B, C, and D) collected from each of the 2 cases. In the other 4 cases, B and C were evaluated.

Sample A: from the mastoid cavity;

Sample B: from the middle ear without performing any manipulation on the cochlea;

Sample C: from the middle ear after opening the round window membrane;

Sample D: from the middle ear after electrode insertion.

We did not collect fluid out of the cochlea since we were interested in the evaluation of the collection procedure.

The study was reviewed and positively evaluated by the Ethical Commission of the Wilhelms Universität Münster (2021-597 fS) and was conducted according to the principles expressed in the Declaration of Helsinki. All of the patients gave their written consent for participation in the study.

## 3. Results

Twelve cochlear implant cases met the surgical indication for revision/resealing of the electrode. There were no patients included under the age of 12.

Depending on the cut-off criteria of CTP values described above, the CTP test was positive in four patients. We observed no intermediate cases (Table 1) in the study group.

The four positive cases were operated on 2 days, 2 months, 3 months, and 2 years after the initial CI surgery, respectively. The mean period after initial surgery for revision was 391 days for the study group. The four CTP-positive cases disclosed rotational vertigo and, in one case, unsteadiness. In the group, seven patients described rotational vertigo, and six described unsteadiness.

Resolution of vertigo symptoms was achieved in one postoperative CTP-positive case (Pat. No. 5). In the three other cases, their rotational vertigo problems were solved, but their unsteadiness persisted. For the whole group, in six other cases, vertigo symptoms decreased after the surgical intervention.

In three of the four CTP-positive cases, the visual assessment was in line with the ELISA finding. In all cases with visual assumption of a PLF, subjective symptoms of vertigo decreased after surgery (six of nine).

The control cases were in line with the clinical expectations of positive and negative findings in terms of CTP testing. A and B samples, as less likely CTP-positive cases (mastoid and middle ear samples), were negative, and C and D, as more likely CTP-positive samples (opening and electrode insertion), were positive cases, if regularly collected. In one case, we observed a high intermediate value. The results were in line with previous studies. The CTP-negative samples A and B, and positive sample D, indicated that the CTP test is accurate in diagnosing PL leakage. Interestingly, a sufficient rinsing of the middle ear (3×) seems to be important for the generation of a positive test in the case of an obvious PLF (Table 2, C).

## 4. Discussion

Cochlear implantation is the treatment of choice for rehabilitating patients with severe-to-profound hearing loss. A quite frequently reported complication after cochlear implantation is postoperative vertigo, found in 13% to 74% of cases [1,2].

In this study, the 12 evaluated patients underwent revisional surgery as a treatment option for their newly occurring vertigo symptoms after cochlear implantation. This treatment has been shown to be a successful surgical option [10], although a psychogenic or placebo effect cannot be fully excluded. CTP is a marker of PLF with high sensitivity and specificity [16] that is used to evaluate the cause of postoperative vertigo objectively.

The most common symptom was rotating vertigo, found in 7 of 12 patients, but six patients also reported unsteadiness. In the first group (CTP positive), all of the patients reported rotational vertigo before the operation. The intraoperative visual test was positive for three of the patients. In the second group (CTP negative), eight patients complained of unsteadiness, and rotational vertigo was documented in two of the patients. The visual test was positive in only three of the patients.

A correlation was not observed between the subjective visual assessment of a fistula and a positive CTP test in this small group, possibly due to the small sample size. However, a relation between the visual assessment and the decrease in symptoms (six positive patients/seven patients with reduced symptoms) can be assumed.

After the surgery, six patients improved significantly, and two of them were symptom free afterwards. Unchanged, persisting vertigo was recorded in three patients.

In one patient (No. 6), barotrauma from flying and climbing was an obvious reason for the appearance of vertigo symptoms. Although CTP testing was negative, this patient improved after surgery. The CT of the patient’s temporal bone showed air inclusion in the labyrinth. A pneumolabyrinth after cochlear implantation is a fairly rare diagnosis. This could be explained by perilymphatic leakage [22,23].

Since this case was CTP negative, we assume that a PLF can temporarily occur and close naturally.

Various reasons for vertigo symptoms after cochlear implantation were discussed, but the major reason was trauma caused by the electrode’s insertion, which can result in an intraoperative loss of perilymph [6]. The most common reasons for a PLF are trauma from head injury, barotrauma, coughing, and sneezing [8]. The occurrence rate of a PLF after cochlear implantation surgery is described as 1% in the literature [24]. The first-line suggested therapies for PLF are medication and vestibular rehabilitation. If the vertigo persists, then an exploratory tympanotomy and packing of the cochleostomy are suggested as treatment options [10,25].

The implantation of an electrode array in the cochlea causes both initial and delayed changes. The initial changes include the trauma caused by the cochlear implantation and disruption of cochlear hemostasis. The tissue response, consisting of inflammation, neofibrosis, and new bone formation, is a change with initial and delayed effects [26]. Importantly, the different sealing techniques used for the cochleostomy significantly influence tissue formation and play an important role because they separate the inner ear compartment from the middle ear [27] to prevent intracochlear infections or leakages that would cause vertigo. Hence, exact sealing of the cochleostomy is crucial. The literature widely discusses several sealing materials, including muscle grafts, carboxylate cement, and no seal at all. The most common sealing graft seems to be autologous fascia or muscle. Sealing with carboxylate cement is controversial because it results in very strong neoplasms of the bone [27]. The material’s absorbability might be or become a problem and increase the likelihood of cochleostomy leakage. Additionally, the material is sensitive to pressure changes. Sneezing may lead to an increase in intracranial pressure, which may result in damage to or breakage of the sealing material. Similarly, pressure changes caused by mountain climbing or flying in an airplane can increase the intratympanic pressure, which might be followed by rupture of the round window membrane or cochleostomy. In any case, an effective barrier should be established as soon as possible after implantation. The fact that the sealing influences the neoplasm of the bone or tissue supports the notion that tissue grows along the electrode array [27].

Different reasons can explain the different prevalence rates between the positive effects of surgery and a positive CTP test. One is a potential placebo effect of the surgery, which affects the relevant psychogenic side of vertigo in some cases. The second might be related to the testing. Although the controls confirmed the method’s specificity, false-negative cases cannot be ruled out completely. A third reason might be a pressure effect during handling of the inserted electrode.

A main limitation of this current study was the lack of objective vestibular tests confirming that the vestibular receptors were affected by the implantation or a possible correlation to a PLF. Additionally, a larger sample size could be used to further underline our findings. Related to the rare occurrence of new vertigo cases after CI surgery, the number of patients is limited. Unfortunately, it was not in all clinically performed revision cases that a middle ear lavage was performed and a sample taken.

## 5. Conclusions

CTP detection objectively showed that PLF occurs in cases of postoperative vertigo after CI.

## Figures and Tables

**Table 1 brainsci-13-01525-t001:** Individual data of implantees of study group.

Pat. Nr.	Prim. Symptom	Time Span CI Revision	Visual Intra OP	CTP	3 Weeks Post OP
2	rot. vertigo	2 days	yes	202	no rotational vertigo but unsteadiness
10	rot. vertigo	2 months	yes	70	no rotational vertigo but unsteadiness
5	rot. vertigo	3 months	no	75	complete recovery
12	rot. vertigo	2 years	yes	80	no rotational vertigo but unsteadiness
6	unsteadiness	1 year	yes	<30	improvement
7	unsteadiness	1 year	yes	<30	improvement
9	unsteadiness	5 years	yes	<30	improvement
1	unsteadiness	12 weeks	no	<30	persisting
11	rot. vertigo	2 months	no	<30	persisting
8	rot. vertigo	8 weeks	no	<30	no rotational vertigobut unsteadiness
3	unsteadiness	2 years	no	<30	persisting
4	rot. vertigo	2 years	no	<30	complete recovery

**Table 2 brainsci-13-01525-t002:** Individual data of controls.

Control	Type of Control	Procedure	CTP Value	Test Results
case 1	A	mastoid probe	1.21	negative
	B	middle ear probe	22.60	negative
	C	probe after RW opening rinsed one time	26.04	negative
	D	probe directly after electrode insertion	148.12	positive
case 2	A	mastoid probe	0.46	negative
	B	middle ear probe	14.13	negative
	C	probe after RW opening rinsed three times	402.47	positive
	D	probe directly after electrode insertion	202.60	positive
case 3	B	middle ear probe	22	negative
	C	probe after RW opening rinsed three times	140	positive
case 4	B	middle ear probe traumatic fistula	82.7	positive
	C	probe after RW opening rinsed three times	172	positive
case 5	B	middle ear probe	11.8	negative
	C	probe after RW opening rinsed three times	223.0	positive
case 6	B	middle ear probe	8.9	negative
	C	probe after RW opening rinsed three times	52.2	intermediate

We studied 6 control patients to verify the correlation between monoclonal CTP ELISA testing and clinical expected results with 4 CI cases. MEL was collected in each of the 4 conditions during CI surgery, with a set of samples (A, B, C, and D) collected from the other 2 cases, B and C: Sample A: from the mastoid cavity. Sample B: from the middle ear without performing any manipulation to the cochlea. Sample C: from the middle ear after opening the round window membrane. Sample D: from the middle ear after electrode insertion. Background color should make a visual differentiation between pos., inter., and neg. cases easier.

## Data Availability

Data are available from the corresponding author upon request.

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
