# Peer review of "Objective Assessment of Perilymphatic Fistula in Cases of Postoperative Vertigo after Cochlear Implantation by Cochlin Tomoprotein (CTP)"

_brainsci, 2023, doi:10.3390/brainsci13111525_

Round 1
Reviewer 1 Report
Comments and Suggestions for Authors
Dear Authors;
Thank you for your effort for this paper. I think you'd better construct the method. First, you have to decide the first purpose.
The authors stated that they conducted the study to evaluate the
positivity rate of the CTP test in cases who developed vertigo after
cochlear implantation. For this purpose, they compared 12 cases who
developed vertigo after CI with 6 cases who underwent CI.
First of all, what is the purpose of this study? Is it to evaluate how
much of the postoperative vertigo in patients undergoing CI is due to
perilymphatic fistula?
Is it to question the value of Cochlin Tomoprotein (CTP) in the
diagnosis of perilymphatic fistula?
CTP positivity was low in patients in the study group (4/12, 33%), and
CTP negative patients also benefited from the revision. CTP was also
found to be high in patients in the control group. No information was
given on whether the patients in the control group had vertigo before or
after surgery. If postoperative vertigo was due to a perilymphatic
fistula, the elevation of CTP would be expected to be higher in the
study group patients. If the patients in the control group had high CTP
but no vertigo (if vertigo were seen, these patients would not have
formed the control group), then the CTP does not support the diagnosis
of perilymphatic fistula for this group of patients. We already know the
role of CTP in the diagnosis of perilymphatic fistula from previous
studies. For these reasons, I think that the editing of the study is not
done well.
In addition, authors should place Table 1 and Table 2 according to where
they are read in the text.
You have to take care when arranging the references (where is reference 7?),
Comments on the Quality of English LanguageMust be revised
Author Response
Dear reviewer,
thank you for your critical observation of our manuscript. Please find a point to point answer to the queries.
Dear Authors;
Thank you for your effort for this paper. I think you'd better construct the method. First, you have to decide the first purpose.
The authors stated that they conducted the study to evaluate the
positivity rate of the CTP test in cases who developed vertigo after
cochlear implantation. For this purpose, they compared 12 cases who
developed vertigo after CI with 6 cases who underwent CI.
First of all, what is the purpose of this study? Is it to evaluate how
much of the postoperative vertigo in patients undergoing CI is due to
perilymphatic fistula?
Is it to question the value of Cochlin Tomoprotein (CTP) in the
diagnosis of perilymphatic fistula?
Control group is to verify/ confirm the value of CTP testing. Patients in the control group were without vertigo. Study group is to evaluate the rate of PLF evidenced by CTP testing in the group of post CI surgery vertigo. We added/ changed the text:
“The aim of this study was to observe the occurrence of middle ear CTP values, as an indicator of PLF, in cases of new occurrences of vertigo after cochlear implantation. Additionally a control group should evaluate the value as a testing for PLF.”
CTP positivity was low in patients in the study group (4/12, 33%), and
CTP negative patients also benefited from the revision. CTP was also
found to be high in patients in the control group. No information was
given on whether the patients in the control group had vertigo before or
after surgery. If postoperative vertigo was due to a perilymphatic
fistula, the elevation of CTP would be expected to be higher in the
study group patients. If the patients in the control group had high CTP
but no vertigo (if vertigo were seen, these patients would not have
formed the control group), then the CTP does not support the diagnosis
of perilymphatic fistula for this group of patients. We already know the
role of CTP in the diagnosis of perilymphatic fistula from previous
studies. For these reasons, I think that the editing of the study is not
done well.
Thank you for this point. We made a control group to underline the evidence of PLF detection ability of CTP testing. Although the rate of PLF/ CTP high in the study group can be seen a slow it is the first objective evidence of PLF in post CI surgery vertigo cases. We added:
“Control group patients were without initial vertigo.“
In addition, authors should place Table 1 and Table 2 according to where
they are read in the text.
Thank you. We change the position of the Tab.
You have to take care when arranging the references (where is reference 7?)
We added reference 7 (L38)
Thank you for your helpful contributions to improve the manuscript.
Reviewer 2 Report
Comments and Suggestions for Authors
Thank you for giving an opportunity to review this paper.
Major points
1. Line 68: Did 12 patients in this study show clinical features of PLF? The authors need to describe about this.
2. In the same context,
Line 91: I think the control group is not suitable for comparison with the study group, because, in the current setting, the authors can only compare the “presence of CTP” between dizzy patients after CI and CI patients just prior to electrode insertion. From the results, little can be answered whether the cause of vertigo or dizziness is PLF or not. It will be better if the authors set control group as “those without dizziness after CI”. Please explain this in Discussion
Minor points
1. Line 21: Cochlea implantation à Cochlear implantation
2. There are lots of typos and grammatical mistakes.
3. Line 44: The authors are recommended to include more recent papers such as, “Surgical Outcomes on Hearing and Vestibular Symptoms in Barotraumatic Perilymphatic Fistula. Ahn J, Son SE, Choi JE, Cho YS, Chung WH. Otol Neurotol. 2019 Apr;40(4): e356-e363. doi: 10.1097/MAO.0000000000002160. PMID: 30870354”.
Comments on the Quality of English LanguageThere are lots of typos and grammatical mistakes.
Author Response
Dear reviewer,
thank you for reviewing the manuscript. Please find a point to point review of the queries.
Thank you for giving an opportunity to review this paper.
Major points
- Line 68: Did 12 patients in this study show clinical features of PLF? The authors need to describe about this.
No, indication for performing a tympanoscopy were described in the method section. Clinical features of PLF (Politzer manoever etc.) are in our experience not suitable in CI patients suffering from vertigo. We add this important information into the method section. “All patients showed no classic clinical features of a PLF.”
- In the same context,
Line 91: I think the control group is not suitable for comparison with the study group, because, in the current setting, the authors can only compare the “presence of CTP” between dizzy patients after CI and CI patients just prior to electrode insertion. From the results, little can be answered whether the cause of vertigo or dizziness is PLF or not. It will be better if the authors set control group as “those without dizziness after CI”. Please explain this in Discussion.
Thank you. The intention of the control group was to verify CTP as an indicator of CTP in our clinical setting. We add this information into the methods section as an information for the control group.” Control group patients were without vertigo.“
Minor points
- Line 21: Cochlea implantation à Cochlear implantation
- Changed
- There are lots of typos and grammatical mistakes.
- The text was re-read by a native speaker
- Line 44: The authors are recommended to include more recent papers such as, “Surgical Outcomes on Hearing and Vestibular Symptoms in Barotraumatic Perilymphatic Fistula. Ahn J, Son SE, Choi JE, Cho YS, Chung WH. Otol Neurotol. 2019 Apr;40(4): e356-e363. doi: 10.1097/MAO.0000000000002160. PMID: 30870354”
- Added
Thank you for the helpful comments to increase the value of the manuscript.
Reviewer 3 Report
Comments and Suggestions for Authors
The authors demonstrated the cause of post operative vertigo after CI by measuring CTP, which is a cochlear-specific protein. The aim of the study is original and interesting, though there are some points to be revised.
1) The research period of the study needs to be clarified, and the number of CI operation, and the percentage of revision after vertigo should be added.
2) The surgical technique of revision surgery is not clear. Please add the explanation.
3) The inclusion criteria of the control cases are not clear, please mention.
4) The age and sex of the cases are not clear, please mention.
5) You should add these points to the limitation section.
Comments on the Quality of English Language
English Proofreading should be performed.
Author Response
Dear reviewer,
thank you for reviewing our manuscript. Please find a point to point answer tot he queries.
The authors demonstrated the cause of post operative vertigo after CI by measuring CTP, which is a cochlear-specific protein. The aim of the study is original and interesting, though there are some points to be revised.
1) The research period of the study needs to be clarified, and the number of CI operation, and the percentage of revision after vertigo should be added.
We added the information. Analysed monoclonal AB was performed between 2021 and 2023. Mean age was 54,5y with 6 female and 6 male implantees. The number of CI surgeries during this period was 221. The total number of revisions was 18. Analysed monoclonal AB sample size was 12.
2) The surgical technique of revision surgery is not clear. Please add the explanation.
The way of performing a revision simple. „Endaural incision, building and elevation of a tymanomeatal flap, identification of the electrode and taking a sample and after that removal oft he tissue patch around the electrode. Placing a new patch around the electrode sealing the middle ear against the inner ear space.“
3) The inclusion criteria of the control cases are not clear, please mention.
The control cases are regular CI cases without vertigo. The samples were taken during different surgial steps. We added the information that the cases are not vertigo cases.
4) The age and sex of the cases are not clear, please mention.
We added the information.
5) You should add these points to the limitation section.
We added the information that the observed number of revision cases is not in line with number of performed revisions with taken samples. We added the number of revision.
Thank you for the helpful comments to improve the manuscript.
Reviewer 4 Report
Comments and Suggestions for Authors
The CTP is a very interesting topic and the work is overall well done, I just would like to clarify some points:
Line 10. Better to write “quite frequent” given the wide variability of the percentages found in literature
Line 20. Better to write “can occur”
Line 27. Better to write “quite frequent”
Line 29. Put “and” between “timely” and “variable”
Lines 30-31. Delete the word “Additional”
Line 54. “PerilympH”
Line 55. “INTraoperative”
Lines 73-74 you said that “resealing was offered as the first treatment option after a vertigo free period”. A) Does this mean that your patients have not received first-line treatments as medication and vestibular rehabilitation for their vertigo? B) If they have spent a vertigo free period, why did you decide to pursue the surgical option?
Lines 70-75 Not clear, write it more clearly and pay attention to full stops and capital letters
Line 87. Delete “the” between from and around
Table 2. “typE”
Line 98. At the end of the line better to write “B and C were evaluated.”
Lines 96-101. This part is not clear. Furthermore you talk about 4 control patients while before you said 6, as in table 2.
Line 119. Not clear why you put table 2 before table 1, it would be better to change the numbers
Line 122. “Of THE study group”
Line 139. Better to write “C and D”
Line 148. Better to write “quite frequently”
Line 165. Delete “of whom”, better to write “2 of them were”
Line 167. Delete the comma
Line 168. Put a comma between “negative” and “this”
Line 180. Delete the “a” between “as” and “treatment options”
Line 195. “Lead to AN INCREASE IN intracranial pressure”
Line 214. Better to write “objectively showed”
Line 218. The T in the first “The” is in bold
1. I would like to better understand the reliability of CTP as a marker of PLF in patient with vertigo after cochlear implant. In your study you said that out of 12 patients only 4 tested positive for CTP, and only in 3 of them you had a visual assessment of PLF, so A) how can you be sure that the other case (CTP-positive) had a fistula? Have you done other investigations such as RM or CT? Furthermore, you also said that the total cases in which you had a visual assumption of PLF were 6, therefore 3 of these, i.e. half, were negative for CTP, B) how do you explain this? With the possibility that the MEL was poorly performed as you said in lines 143-144? C) And how can you be sure that other patients, CTP-negative, with no visual assessment didn't have PLF?
2. Finally, if the validity of CTP as a marker in patients with post-cochlear implant vertigo were further confirmed, how do you think this discovery could be used to improve the intervention attitude in daily practice?
As you said, a larger sample size could add validity to your findings but is not simple since new vertigo cases after CI surgery are rare. I will follow your findings with interest.
Author Response
Dear reviewer,
thank you for reviewing the manuscript. Please find a point to point answering of the queries.
The CTP is a very interesting topic and the work is overall well done, I just would like to clarify some points:
Line 10. Better to write “quite frequent” given the wide variability of the percentages found in literature
changed
Line 20. Better to write “can occur”
changed
Line 27. Better to write “quite frequent”
changed
Line 29. Put “and” between “timely” and “variable”
changed
Lines 30-31. Delete the word “Additional”
changed
Line 54. “PerilympH”
changed
Line 55. “INTraoperative”
changed
Lines 73-74 you said that “resealing was offered as the first treatment option after a vertigo free period”. A) Does this mean that your patients have not received first-line treatments as medication and vestibular rehabilitation for their vertigo?
You are right. We added: …that still persisted unchanged after 4 days of steroid and antibiotic medication… We removed „first treatment option“.
- B) If they have spent a vertigo free period, why did you decide to pursue the surgical option?
We assume that after a period of tight sealing, a patch can shrink or can be absorbed like we know that from some cases of tympanoplasty.
Lines 70-75 Not clear, write it more clearly and pay attention to full stops and capital letters
Line 87. Delete “the” between from and around
changed
Table 2. “typE”
unclear
Line 98. At the end of the line better to write “B and C were evaluated.”
changed
Lines 96-101. This part is not clear. Furthermore you talk about 4 control patients while before you said 6, as in table 2.
Thank you. 6 patients is right.
Line 119. Not clear why you put table 2 before table 1, it would be better to change the numbers
Tab.1 and Tab.2 position is changed
Line 122. “Of THE study group”
changed
Line 139. Better to write “C and D”
changed
Line 148. Better to write “quite frequently”
changed
Line 165. Delete “of whom”, better to write “2 of them were”
changed
Line 167. Delete the comma
done
Line 168. Put a comma between “negative” and “this”
done
Line 180. Delete the “a” between “as” and “treatment options”
done
Line 195. “Lead to AN INCREASE IN intracranial pressure”
done
Line 214. Better to write “objectively showed”
changed
Line 218. The T in the first “The” is in bold
changed
- I would like to better understand the reliability of CTP as a marker of PLF in patient with vertigo after cochlear implant. In your study you said that out of 12 patients only 4 tested positive for CTP, and only in 3 of them you had a visual assessment of PLF, so A) how can you be sure that the other case (CTP-positive) had a fistula? Have you done other investigations such as RM or CT? Furthermore, you also said that the total cases in which you had a visual assumption of PLF were 6, therefore 3 of these, i.e. half, were negative for CTP, B) how do you explain this? With the possibility that the MEL was poorly performed as you said in lines 143-144? C) And how can you be sure that other patients, CTP-negative, with no visual assessment didn't have PLF?
To A: CTP testing is a reliable tool for PLF evaluation confirmed by various literature and by our own controls. We have not performed a CT.
To B: Visual assessment is reliable only to a limited degree. A poor performance of the testing is a possible factor to generate false negative results. That is right.
To C: I assume that a PLF case with a negative testing would at least be in the intermediate value (30-60 ng) results group. This is not the case in our study group. But you are absolutely right. How to take the sample is a very important point.
- Finally, if the validity of CTP as a marker in patients with post-cochlear implant vertigo were further confirmed, how do you think this discovery could be used to improve the intervention attitude in daily practice?
With increasing acceptance re-sealing procedures will be performed frequently in CI surgery centers. Re-sealing is a low risk procedure with a high effect on new vertigo disabled patients after CI surgery.
As you said, a larger sample size could add validity to your findings but is not simple since new vertigo cases after CI surgery are rare. I will follow your findings with interest.
Thank you for your interest and support to improve the quality of the manuscript.
Round 2
Reviewer 1 Report
Comments and Suggestions for Authors
-
Author Response
Thank you !
Reviewer 3 Report
Comments and Suggestions for Authors
I consider that the manuscript has been sufficiently improved to warrant publication in Brain Sciences.
Author Response
Thank you !